# Building knowledge, optimising physical and mental health and setting up healthier life trajectories in South African women (*Bukhali*): a preconception randomised control trial part of the Healthy Life Trajectories Initiative (HeLTI)

Shane A Norris ,[1,2] Catherine E Draper,[1] Alessandra Prioreschi,[1] CM Smuts,[3] Lisa Jayne Ware ,[1,4] CindyLee Dennis ,[5] Philip Awadalla,[6] D Bassani,[7] Zulfiqar Bhutta ,[7,8] Laurent Briollais,[9] D William Cameron,[10,11] Tobias Chirwa,[12] B Fallon,[13] CM Gray,[14,15] Jill Hamilton ,[16] J Jamison,[17] Heather Jaspan,[15] Jennifer Jenkins,[18] Kathleen Kahn,[12,19] AP Kengne,[20] Estelle V Lambert,[21] Naomi Levitt,[22] Marie-Claude Martin,[9] Michele Ramsay,[23] Daniel Roth,[16,24] Stephen Scherer,[24,25] Daniel Sellen,[26] Wiedaad Slemming ,[27] Deborah Sloboda,[28] M Szyf,[29] Stephen Tollman,[12,19] Mark Tomlinson ,[30] Suzanne Tough,[31] Stephen G Matthews,[9,32] Linda Richter,[4] Stephen Lye[1,9,32]

**Correspondence to**
Dr Shane A Norris;
shane.norris@wits.ac.za

## ABSTRACT

**Introduction** South Africa's evolving burden of disease is challenging due to a persistent infectious disease, burgeoning obesity, most notably among women and rising rates of non-communicable diseases (NCDs). With two thirds of women presenting at their first antenatal visit either overweight or obese in urban South Africa (SA), the preconception period is an opportunity to optimise health and offset transgenerational risk of both obesity and NCDs.

**Methods and analysis** *Bukhali* is the first individual randomised controlled trial in Africa to test the efficacy of a complex continuum of care intervention and forms part of the Healthy Life Trajectories Initiative (HeLTI) consortium implementing harmonised trials in Canada, China, India and SA. Starting preconception and continuing through pregnancy, infancy and childhood, the intervention is designed to improve nutrition, physical and mental health and health behaviours of South African women to offset obesity-risk (adiposity) in their offspring. Women aged 18–28 years (n=6800) will be recruited from Soweto, an urban-poor area of Johannesburg. The primary outcome is dual-energy X-ray absorptiometry derived fat mass index (fat mass divided by height$^2$) in the offspring at age 5 years. Community health workers will deliver the intervention randomly to half the cohort by providing health literacy material, dispensing a multimicronutrient supplement, providing health services and feedback, and facilitating behaviour change support sessions to optimise: (1) nutrition, (2) physical and mental health and (3) lay

## Strengths and limitations of this study

► *Bukhali* will be the first randomised controlled trial in Africa to evaluate if a community health worker facilitated intervention starting preconception will reduce offspring obesity risk at age 5 years.

► Primary and many secondary outcomes are harmonised with three other Healthy Life Trajectories Initiative trial sites (Canada, China and India) to enable comparative and pooled analyses.

► The impact of *Bukhali's* data and learning will be to inform preconception prevention of non-communicable disease risk strategies in Africa.

► A limitation is that *Bukhali* does not include an economic evaluation.

the foundations for healthier pregnancies and early child development.

**Ethics and dissemination** Ethical approval has been obtained from the Human Ethics Research Committee University of the Witwatersrand, Johannesburg, South Africa (M1811111), the University of Toronto, Canada (19-0066-E) and the WHO Ethics Committee (ERC.0003328). Data and biological sample sharing policies are consistent with the governance policy of the HeLTI Consortium (https://helti.org) and South African government legislation (POPIA). The recruitment and research team will obtain informed consent.

**Trial registration** This trial is registered with the Pan African Clinical Trials Registry (https://pactr.samrc.ac.za) on 25 March 2019 (identifier: PACTR201903750173871).

**Protocol version** 20 March 2022 (version #4). Any protocol amendments will be communicated to investigators, Institutional Review Board (IRB)s, trial participants and trial registries.

## BACKGROUND

Non-communicable diseases (NCDs) such as obesity, hypertension and diabetes are significant health burdens in many rapidly transitioning middle-income countries.[1] NCDs kill 41 million people each year (71% of deaths) globally, and WHO has identified NCDs as one of the 10 leading threats to health.[2] From 1980 to 2014, the prevalence of diabetes changed little in western Europe, but doubled in sub-Saharan Africa (SSA), India and China.[1] SSA countries have persisting afflictions from infectious diseases (HIV, Tuberculosis (TB) and malaria) against a backdrop of maternal and child malnutrition,[3] but also now the COVID-19 pandemic. In many respects the impetus to address NCDs as part of the Sustainable Development Goals has stalled as a result of the COVID-19 emergency.

Childhood weight gain, overweight and obesity (OWO) and adiposity are risk factors for poor health trajectories and development of NCDs. The number of children and adolescents with obesity has risen to 124 million worldwide and could exceed 250 million by 2030 without substantial interventions.[4] A meta-analysis of 61 studies showed that increased preconception and interpregnancy weight gain was associated with gestational diabetes mellitus (GDM; 88% greater risk), pre-eclampsia (92% greater risk) and large-for-gestational-age (36% greater risk) babies.[5] A systematic review of 38 studies found a consistent association between maternal prepregnancy weight and offspring obesity.[6] Furthermore, Heslehurst and colleagues showed that a child has a 2.6-fold greater risk of obesity if the mother is obese prior to conception.[7] A systematic review by Dean and coauthors identified 23 randomised controlled trials of preconception care and counselling delivered via health facilities or community groups, that reported improved women's health behaviours (calorie restriction and increased physical activity), as well as, more positive maternal, neonatal and child health outcomes including an increase in antenatal care seeking and an increased likelihood of breast feeding.[8 9] None of these preconception interventions were conducted in Africa.

South Africa's (SA) evolving burden of disease is complex due to: (1) persisting malnutrition; (2) the highest prevalence of OWO in SSA, particularly among women; and (3) rising rates of NCDs such as type 2 diabetes. In Soweto (urban-poor area of the Johannesburg Metropole covering 200 km² with over 1.3 million residents; 6400/km²), we have found that the prevalence of OWO in females increases from 10% by age 8% to 43% by age 22 years, while the prevalence in males remains fairly constant into young adulthood at 10%; further, girls who are obese by age 5 are 42 times more likely to be obese adults.[10 11] Furthermore, 66% of pregnant Soweto women at first antenatal clinic visit were either overweight or obese, and 10% were diagnosed with GDM in late pregnancy.[12 13] Consequently, there is a high obesity burden need in Soweto for preconception interventions to support women to optimise health and manage weight to combat intergenerational obesity.[14]

The Healthy Life Trajectories Initiative (HeLTI), an international consortium developed in partnership with the WHO to address childhood obesity. We hypothesise that an integrated complex intervention comprising a continuum of care starting preconception and across pregnancy, infancy and childhood will reduce childhood adiposity and the risk for NCDs, as well as, improve child development. Four randomised controlled trials in Ontario and Alberta (Canada),[15] Shanghai (China),[16] Mysore (India)[17] and Soweto (South Africa) have been harmonised to test this hypothesis. These sites were selected based on the community need and that they possessed both the infrastructure, and multidisciplinary teams with cohort/trial experience to execute an ambitious 10-year programme of work. This paper describes the HeLTI South African trial protocol in Soweto called *Bukhali* (a Zulu word meaning smart/powerful) signifying the beneficial potential of optimising women's health for herself and her future offspring.

## METHODS/DESIGN

### Patient and public involvement

We constituted a Community Advisory Group with local women, and have involved them in the intervention development and the preparatory phase to enable feedback and input into the feasibility of the study. Furthermore, a Participant Advisory Group has been formed as part of a longitudinal qualitative study to further understand and identify preconception health concerns, barriers and solutions, as well as, other contextual issues that participants are facing which could influence their participation in the trial. Lastly, a Stakeholder Advisory Group has been formed with multiple stakeholders (South African government departments of health and education, UNICEF, Save the Children, WHO, Non-Government Organisations (NGOs) and academics) to share the development and results of *Bukhali* and elicit input and guidance around a variety of issues—policy relevance, linkage with current initiatives, ethics and dissemination.

### Formative research and pilot trial study

The trial was designed following the Standard Protocol Items (Recommendations for Interventional Trials (SPIRIT) 2013 statement). Through extensive qualitative (focus groups, in-depth individual interviews and stakeholder engagement)[18–23] and epidemiological research,[24–31] we identified our young women (18–28 years) as the target group given that a high proportion these women are already either overweight or obese and

most will have their first child during this age range, and developed the intervention package. We initially opted for a cluster randomised study design in order to included community peer group sessions as part of the intervention. To assess the feasibility and implementation of this design, a pilot trial of data collection and intervention commenced in 2018 and concluded in 2019. We recruited 1719 participants within six community clusters, randomised the clusters, collected baseline data and biological samples, and implemented the planned intervention and control arm activities. The pilot trial data highlighted critical learning, among others, that: (1) it was feasible to recruit and collect good quality data and biological samples (95% consented to provide a blood sample and 75% consented to provide a DNA sample) from a large group of our target population; (2) young women living in Soweto lead complicated and economically stressful lives and are preoccupied with survival strategies, which often entails moving and living at multiple homes across family and friends in the community over time; (3) there is poor health literacy around preconception health among young women in Soweto; (4) young women were anxious about participation due to fear of criminal activities and abduction; (5) community safety of study community health workers (CHWs) could not be assured; and (6) we could not effectively deliver the planned community peer group component due to poor attendance despite convenient Saturday meeting times, multiple opportunities during the month to attend, and the provision of refreshments. To address these challenges, we formulated a multipronged strategy: (1) we rolled-out a communication campaign; a soap opera series of 11 episodes drawing attention to preconception health issues in multiple languages through two local radio stations with the largest population reach supported by social media engagement (https://soundcloud.com/user-662329775); (2) we engaged and obtained additional support to operate within the community through community leadership forums and police awareness of the study team; (3) we produced branded material for staff and vehicles with clear university and project identification to promote credibility; and (4) we modified the study design (we pivoted from cluster randomisation to individual randomisation due to cluster contamination linked to young women shifting between multiple households) and intervention delivery, and jettisoned the peer group component because of poor compliance.[32]

## Study design

*Bukhali* is an individual-randomised controlled two-arm trial (simple randomisation; 1:1 ratio to the intervention arm or to standard of care plus control arm). After informed consent and baseline data collection, the participant moves to a separate/private randomisation counter where the participant clicks on a computer-generated electronic envelope to reveal the arm allocation. The randomisation officer then escorts the participant to a separate wing on a different floor within the centre to introduce participant to either the intervention arm or control arm coordinator, thus preserving the concealment of allocation. The research trial data team and associated researchers will be blinded to participant allocation throughout the trial, and this will be achieved by: (1) ensuring that group allocation is locked on the database and cannot be accessed by the research team; (2) training the research team not to ask and encouraging participants not to divulge their allocation status; and (3) when participants visit the centre for their respective arm programme these activities will occur in a separate part of the centre away from the research team. All participants will be exposed to the preconception intervention or control programme for up to 18 months. If a participant does not become pregnant within the 18 months, she will have an exit assessment. If a participant becomes pregnant, she will exit the preconception phase and be monitored through pregnancy, and postnatally. An 'index child' conceived after randomisation will be followed until age 5 years. Participant flow through the trial is seen in figure 1.

## Inclusion/exclusion criteria

Participants will be recruited through a survey of Soweto households (approximately 30 000), which is sufficiently large to adequately recruit our target participant number. Women aged 18–28 years are eligible, unless they are: (1) diagnosed with type 1 diabetes or epilepsy because these require intensive treatment and management priorities; (2) present with intellectual disability that hinders informed consent and (3) not able or willing to provide consent. Pregnancy at the time of the baseline assessment does not exclude participation. Participants who move residence during the course of the study will continue participation as long as their current address is known, the participant wishes to remain involved, and data can be collected. Participants who have miscarriages will be offered counselling support and remain in the study.

## Sample size

The sample size calculation is based on the mean (4.1 kg/m$^2$) and SD (2.1) for dual-energy X-ray absorptiometry (DXA) derived fat mass index (FMI) calculated from our prepubertal data from the Birth to Twenty Plus Cohort in Soweto, South Africa. We will require 765 children to be born in each arm to detect a meaningful difference of 0.25 SD (0.5 kg/m$^2$) in FMI at age 5 years between the intervention and control groups at 80% power and 5% significance level, and assuming 30% attrition. Therefore, we will recruit approximately 6800 women so reach a target of 1530 pregnancies. The intervention group will receive all four phases of the intervention and the control remains standard of care plus throughout. Pooling of data across the four countries will provide the ability to detect an effect size of 0.1 SD between intervention and control groups at 80% power and 5% significance level. The pooled analysis should also allow us to detect a 0.2 SD difference in our primary outcome in boys and girls

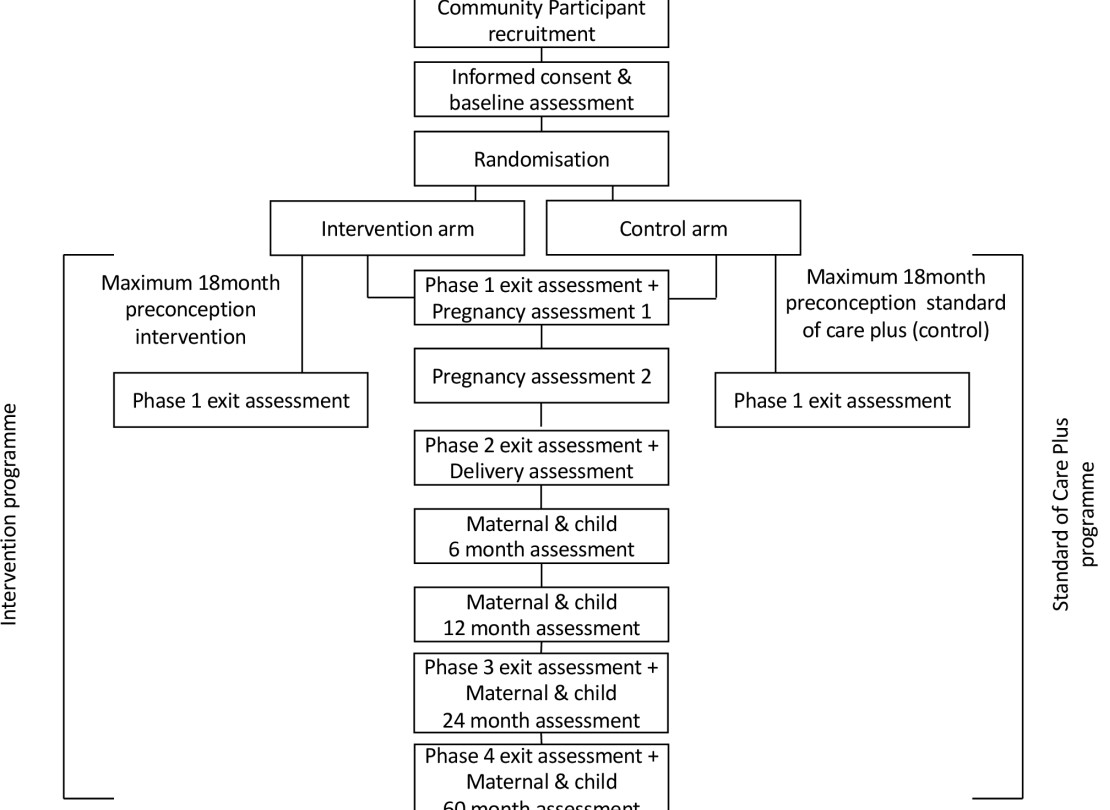

**Figure 1** Participant flow diagram.

separately (although this is dependent on the direction of change being the same across our populations).

### Primary and secondary outcome questions

Primary child outcome: What is the relative effect of a 4-phase intervention on adiposity status at age 5 years as determined by DXA-derived FMI (fat mass/height$^2$ among children of young adult women in the intervention arm, when compared with the children of women who receive standard of care plus (control arm)?

Secondary child outcomes: What are the effects of the 4-phase intervention on child outcomes at age 5 years on:

1. Anthropometry (age-standardised and sex-standardised continuous body mass index (BMI)-z score, BMI growth trajectories, continuous waist circumference z score, upper arm circumference, head circumference);
2. Cardiometabolic outcomes (systolic and diastolic blood pressure, lipid profile, glucose, insulin, insulin resistance);
3. Behaviour (dietary intake, physical activity, screen time, sedentary behaviour, sleep time);
4. Developmental outcomes (cognitive, motor skills, communication, behavioural); and

Secondary maternal outcomes: What are the effects of the 4-phase intervention on?

1. Anthropometry (OWO, DXA-derived body composition, waist circumference);
2. Nutrition (anaemia)
3. Physical health (hypertension);

4. Mental health (depressive symptoms, anxiety) and
5. Behaviour (dietary patterns, physical activity, screen time, sedentary behaviour, sleep time, tobacco and alcohol use);

### Phase-Sspecific outcomes

1. Preconception Phase (outcome at conception or exit): What is the effect of the preconception intervention phase on—maternal anthropometry and body composition (OWO, FMI, waist circumference); nutrition (anaemia); health behaviour (physical activity, screen time, sleep); mental health (depressive symptoms, anxiety); and physical health (hypertension)?
2. Pregnancy Phase (parental and child outcomes at birth): What is the effect of the preconception + pregnancy intervention phases on—pregnancy weight gain; rate of GDM; rate of gestational hypertension; rate of pre-eclampsia; rate of preterm birth <37 weeks; rates of large and small birth weight for gestational age; and maternal mental health?
3. Infancy Phase (parental and child outcomes across the first 2 years post partum): What is the effect of the preconception + pregnancy + infancy intervention phases on: breastfeeding outcomes (initiation, duration, exclusivity, self-efficacy); nutrition (anaemia), health behaviour outcomes (physical activity, screen time, sleep); and mental health outcomes (depressive symptoms, anxiety) and child outcomes (growth, blood pressure, sleep, sedentary and physical activity; feeding practices; developmental outcomes)?

4. Early Childhood phase (parental and child outcomes 2–5 years post partum): What is the effect of the preconception + pregnancy + infancy + childhood intervention phases on: maternal nutrition (anaemia), health behaviour outcomes (physical activity, screen time, sleep); and mental health outcomes (depressive symptoms, anxiety); and child outcomes (growth, blood pressure, sleep, sedentary and physical activity; dietary practices; cognitive development outcomes)?

### Ethics and regulatory management

*Bukhali* has ethical approval from the Human Ethics Research Committee of the University of the Witwatersrand, Johannesburg, South Africa (M1811111), the University of Toronto, Canada (19-0066-E) and the WHO Ethics Committee (ERC.0003328). The trial is registered with the Pan African Clinical Trial Registry.

### Intervention: CHW approach

The intervention has been informed by the UK MRC Guidelines for Complex Interventions, and we used the TIDieR (Template for Intervention Description and Replication) Checklist. Using the taxonomy of behaviour change techniques,[33] we grounded the behaviour change intervention in: (1) theory of planned behaviour, which suggests that behaviours are determined by attitudes, subjective norms and perceived control; (2) control theory, which suggests we use inner-control and outer-control to minimise deviation from the standard; and (3) social cognitive theory, which posits behaviour and

choices are influenced by the environment, personality and other factors such as goals. These theories combined anchors the intervention to value the importance of context, individual needs and support to bolster agency. Figure 2 details the original logic model to illustrate the pathways to impact that guided the intervention development. The intervention is centred on 'Living your best life' with four universal objectives that infuse through all phases of the intervention (preconception, pregnancy, infancy and childhood) to: (1) optimise health; (2) optimise nutrition; (3) optimise mental health; and (4) promote early childhood development. The agents of change are modelled on CHWs, who we call 'Health Helpers' (HHs) to differentiate them from current Department of Health workers. We opted for the CHW approach as there is positive efficacy evidence in Africa, and it aligns with SA's primary healthcare re-engineering policy. The HHs will share similar qualifications, characteristics and salary levels to CHWs in SA, and eligibility includes women between the ages of 23 and 40 years with completed secondary school with some tertiary education, but no formal training as a CHW is required.

The HHs are supported by a supervisor and dietician with access to nurses and a referral network. HHs deliver the intervention through the provision of health literacy resource materials and micronutrient supplements, provide health feedback and services, and they support behaviour change through individual sessions in person or telephonically, and supportive text messaging. Multiple

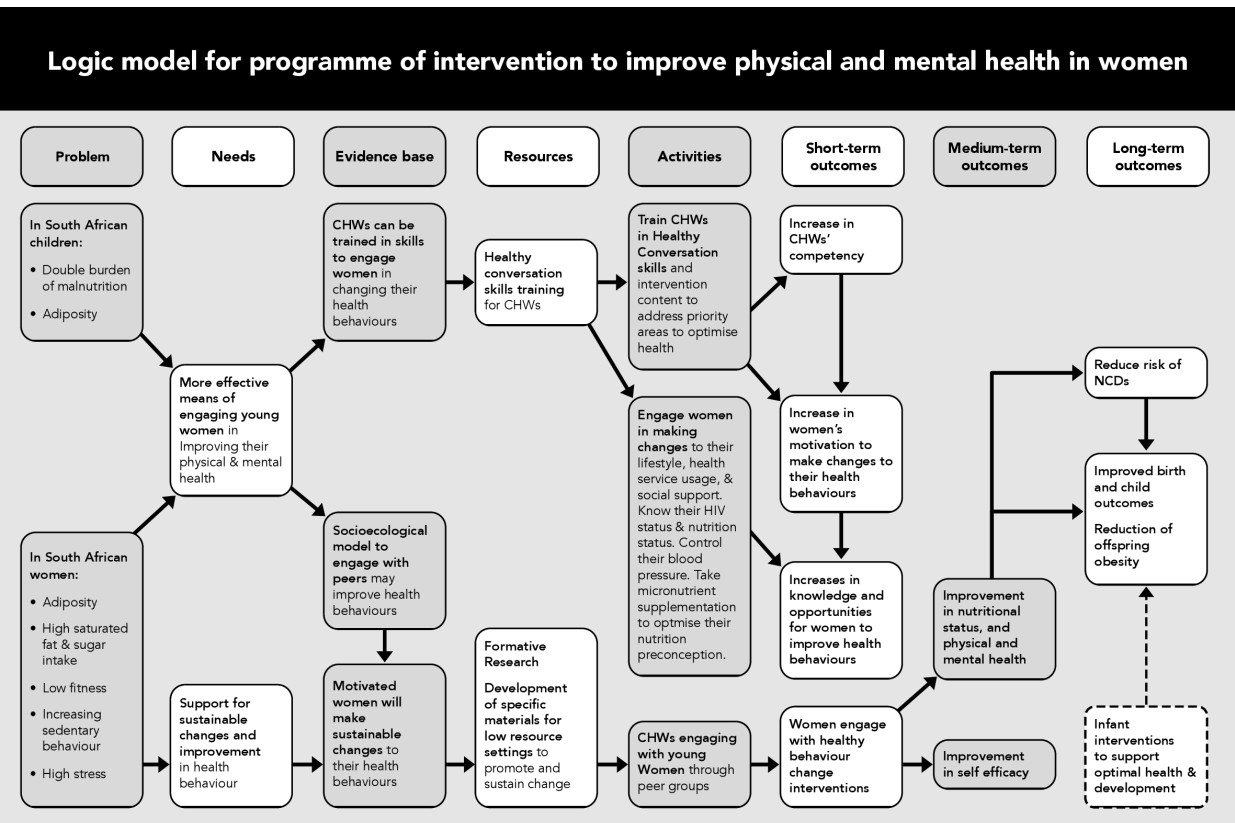

**Figure 2** Pathways to impact model that informed the intervention development. CHWs, community health workers

strategies will be put in place by the HHs (flexibility, reminders, home visits, incentives) to aid in achieving trial compliance.

### Healthy conversation skills and trauma-informed care

HHs will be trained in healthy conversation skills (HCS), a technique specifically developed and tested for use with socioeconomically disadvantaged participants to improve self-efficacy and support behaviour change. HCS is based on a social-cognitive model of health behaviour, which emphasises the role of increasing self-efficacy in promoting behaviour change.[34–36] It recognises that knowledge alone is insufficient to change behaviour unless people are also motivated and empowered to change. Trained in HCS, HHs will explore the barriers that women experience in trying to change their behaviour, ways of overcoming these barriers and setting goals for change. The HCS approach enables a woman to reflect on her particular priorities for physical and mental health, and to support her in finding her own solutions to challenges she experiences. The training is standardised and supported by the MRC Lifecourse Epidemiology Unit (UK). HCS training is aimed at achieving four basic competencies that are known to be useful in supporting behaviour change: (1) the use of open-discovery questions, questions that begin with 'how' and 'what' and lead people to explore and find their own ways of overcoming barriers to change; (2) listening more than talking which allows people to identify and take control of their own behaviour change; (3) reflecting on practice in order to be more effective; and (4) supporting the use of SMARTER goal setting so that participants have a sense of change and progress. To the HCS approach, we have added selected skills from the WHO Problem Management Plus (individual psychological help for adults impaired by distress in communities exposed to adversity).

### Health literacy resource material

With topic experts and a specialised curriculum developer, we integrated behaviour change theory and designed both HH facilitator manuals and participant resource materials. The resource material encompasses nine modules (women's health, chronic diseases, diet, sleep, physical activity and fitness, sitting time, body size and image, emotional awareness and caregiving) in six chapters, and these modules repeat with new content every six chapters. The resource books contain essential educational content, local narratives to convey key concepts and space for participants to engage with the material through the use of checklists, diaries and reflections. The approach for each module is that of: (1) Knowing (what do I know about it?); (2) Doing (what am I doing about it?); and (3) Becoming (how do I live my best life?). Three preconception health resource books are provided; one book is provided for pregnancy health, and supplementary resource materials to the South African Road to Health Booklet (https://sidebyside.co.

**Table 1** Nutrient composition of multiple micronutrient supplements

| Micronutrient | Unit | NRV EU |
|---|---|---|
| Vitamin A | 2664 IU | 100% (800 µg/2664 IU) |
| Vitamin D$_3$ | 200 IU | 100% (5 µg/200 IU) |
| Tocopherol (vitamin E) | 15 mg | 125% (12 mg) |
| Vitamin K1 | 55 µg | 73% (75 µg) |
| Thiamine | 1.4 mg | 127% (1.1 mg) |
| Riboflavin | 1.4 mg | 100% (1.4 mg) |
| Niacinamide | 18 mg | 113% (16 mg) |
| Pantothenic acid | 6 mg | 100% (6 mg) |
| Pyridoxine (vitamin B$_6$) | 1.9 mg | 136% (1.4 mg) |
| Biotin | 30 µg | 60% (50 µg) |
| Folic acid | 600 µg | 300% (200 µg) |
| Cyanocobalamin (vitamin B$_{12}$) | 2.6 µg | 100% (2.5 µg) |
| Ascorbic acid | 60 mg | 75% (80 mg) |
| Copper | 1.15 mg | 100% (1 mg) |
| Iodine | 250 µg | 167% (150 µg) |
| Iron | 27 mg | 193% (14 mg) |
| Selenium | 50 µg | 100% (55 µg) |
| Zinc | 10 mg | 100% (10 mg) |

NRV, nutrient reference value.

za/resources/road-to-health-book/) and nurturing care resources are provided for infancy and early childhood.

### Multimicronutrient supplementation

Based on 6-monthly point-of-care haemoglobin concentration assessments, the HHs will dispense and monitor micronutrient supplement use. The multimicronutrient (MMN) supplement is based on WHO recommendations (table 1) to optimise nutrition and address anaemia, based on haemoglobin concentrations:
▶ Non-anaemic women (Hb ≥12 g/dL at enrolment): provision of a micronutrient supplement containing approximately 27 mg iron twice (2×) per week.
▶ Mildly anaemic women (Hb <12 g/dL and >7 g/dL at enrolment): Provision of a micronutrient supplement daily for 6 months.
▶ Severely anaemic women (Hb <7 g/dL at baseline): receive treatment regime according to current standard of care in South Africa.

Supplement use will continue throughout the preconception period until pregnancy or exit. If the participant becomes pregnant, the supplement regimen will then shift to daily use throughout pregnancy and 6 months post partum.

### Services

Through individual in-person or telephonic sessions, HHs will provide: (1) specific health feedback to participants based on their baseline results (BMI (obesity), blood

pressure (hypertension), haemoglobin concentration (anaemia), mental health and lifestyle (physical activity, sedentary behaviour, sleep)); (2) offer home-based HIV and pregnancy testing with counselling, use HCS to identify aspects of health behaviour change, set goals and monitor and provide access to healthcare referral networks. For overweight and obese women, additional waist circumference measurements will be provided. During the infant and child periods, HHs continue the monthly consultations and support women with aid of the South African Road to Health Booklet and postnatal clinic visits.

### Control arm: call centre approach (standard of care plus)

In addition to access to standard of care offered through the public health system, control arm participants will have a dedicated call centre. The rationale to add a standard of care plus programme is to maintain contact with participants, and minimise attrition bias. Similar to the intervention team, the call centre team will receive training in the delivery of the modules. To partially control for special attention given to women in the intervention arm, the call centre will contact participants once a month and deliver a curriculum on 'life skills' via telephone, SMS and email. The non-health specific intervention material will be standardised and the modules of the curriculum will include: (1) civic rights and responsibilities (How do I apply for an identity document; what does it mean to vote?); (2) financial education (savings and budgets; how do I open a bank account); (3) job readiness (preparing a curriculum vitae (CV); job interview skills; volunteering); (4) navigating social media (internet, different social media; how social media can be harmful) and (5) social grants. HIV and pregnancy testing services will be offered on request for free at the research centre.

### Process evaluation

For all intervention (health worker-centric) and control (call centre-centric) components, standard operating procedures (SOPs) support the HHs and call centre team to deliver their respective modules in a structured and uniform way. We will follow the MRC guidance on process evaluation of complex interventions.[37] Evaluation will focus on implementation, mechanisms of impact and context. Assessment of implementation will focus on finding out what is delivered and how delivery is achieved, and which is vital information if the intervention is to be replicated. We will assess compliance by counts using a log (eg, contacts made), and by discussions between HHs and participants. We will measure fidelity of intervention delivery by observing HHs during one-to-one contacts with intervention group participants, focusing on HHs use of HCS and the competencies they demonstrate. We will also observe the contact between call centre assistants and control participants to record the interactions. We will monitor intervention dose by recording the frequency and duration of contact between participants and HHs using diaries. To understand the mechanisms through which the intervention brings about change in the trial's specified outcomes, we will assess the acceptability of the intervention,

and women's experiences of contact with HHs using interviews and focus groups. In assessing the context, we will aim to identify factors that might act as barriers or facilitators to intervention implementation or effects. This will include local and national policy, local service configuration and provision and sociodemographic and environmental factors within the communities taking part in the study. Qualitative methods (interviews and focus groups), including longitudinal approaches, will be used to further explore specific issues of relevance, in terms of participants' experiences of the trial, key issues facing participants and factors influencing implementation of the trial.

### Data collection, management and analysis

Tables 2 and 3 details the phenotype measures and biological sample collection on women, biological fathers and index child. Formative research has indicated low-level (10%) biological father participation in the study. A dedicated research team will be established to increase the father participation through alternate approaches (eg: data collection flexibility in evening and on Saturdays; encouraging fathers to accompany pregnant women for their ultrasound scan). There is substantial harmonisation of data, samples and quality management across the four HeLTI studies. For example, over 1800 core variables have been harmonised from preconception through to infancy (6 months). All data will be managed through real-time capturing, reporting and quality control utilising REDCap electronic data capture tools.[38] Maelstrom Research at McGill University (Canada) has been supporting the harmonisation process across the sites and developing the online catalogue and metadata, which are available on their website (https://www.maelstrom-research.org/study/helti-hp). In addition, Maelstrom will support the implementation of a data infrastructure within each country using DataSHIELD/Opal software[39] to allow federated data analysis (country-specific or meta-analysis by pooling estimates across countries) to be conducted without having access to individual participant data or exporting and pooling data on a central server. This capacity will provide a mechanism for addressing restrictive regulation regarding exporting data across countries. The data management system coupled with our laboratory management system of samples (different tissues at different time points) will enable researchers to conduct future mechanistic studies and participants have provided consent for this on provision of IRB approval for such studies. All personal data are managed separately and secured with restricted access in line with the Protection of Personal Information Act of South Africa.

### Analysis plan

Standard data screening/cleaning procedures will be conducted including the extent and pattern of missing data. When data are missing at random, which we expect to be the case, we will use multiple imputation so as to maximise sample size and study power in a dataset. We will assess the comparability at randomisation of the groups using descriptive analyses. We will also assess process indicators including recruitment and retention,

**Table 2** Longitudinal data collection measures

| Domain | Measure | Preconception | | Pregnancy (week) | | Delivery | Postnatal period (months) | | | |
|---|---|---|---|---|---|---|---|---|---|---|
| Time point (months) | | Baseline | 18 months (exit) | 10–17 | 24–28 | Delivery | 6 | 12 | 24 | 60 |
| Demography | Demographic and socioeconomic data, household composition, occupation, education, living environment | M | M | M | F | | M | M | M | M |
| | Date of birth/sex | M | | | F | C | C | C | C | C |
| Anthropometry and body composition | Height/length | M | | M | F | C | C | C | C | C |
| | Weight | M | M | M | M/F | C | M/C | M/C | M/C | M/C |
| | Gestational weight gain | | | M | M | | | | | |
| | Body mass index | M | M | M | M/F | | M | M | M | M |
| | Adiposity: DXA (adult) | M | M | | F | | M | | M/C | M/C |
| | Skinfolds (triceps and subscapular) (*if scan is not possible) | M | | M | M | | M* | | M* | M* |
| | Peapod body composition | | | | | C | C | | | |
| | Waist circumference | M | M | | F | C | C | M/C | M/C | M/C |
| | Fetal anthropometry (gestational age and fetal growth) | | | C | C | C (GA) | | | | |
| | Mid upper arm circumference | M | M | M | M/F | C | C | C | M/C | M/C |
| | Head circumference | | | | | C | C | C | C | C |
| Clinical | Pregnancy complications | | | M | M | M/C | C | C | C | C |
| | Birth complications, survival | | | | | C | | | | |
| | APGAR score 1 and 5 mins | | | | | C | | | | |
| | HIV test | M | M | M | F | | | | M | M |
| | Pregnancy test (pre-DXA scan) | M | M | M | | | M | | M | M |
| | Blood pressure | M | M | M | M/F | | M | M | M/C | M/C |
| | Anaemia (HemoCue) | M | M | M | M/F | | M | M | M | M |
| | HbA1c | M | M | M | M/F | | | M | M | M |
| | Random blood glucose | M | M | M | F | | | | | |
| | Fasting blood glucose | | | | M | | | | | |
| | Oral glucose tolerance test | | | | M | | | | | |
| | Plasma and serum collection | M | M | M | M/F | | | M | M/C | M/C |
| | DNA collection | M | M | M | M/F | | | M | M/C | M/C |
| | Urine spot sample | M | M | M | M/F | | | | C | M/C |
| | Urine dipstick | M | M | M | M/F | | | | C | M/C |
| | Heel-prick blood | | | | M/F | | | | C | |
| | Buccal swab | | | | M/F | | | | C | |
| Lifestyle and health behaviour | Diet intake (Dietary Diversity and Household Food Insecurity Access Scale) | M | M | M | M/F | | | M | M | M |
| | Breastfeeding Questionnaire | | | | | M | | | | |

**Table 2** Continued

| Domain | Measure | Preconception Baseline | 18 months (exit) | Pregnancy (week) 10–17 | Pregnancy (week) 24–28 | Delivery | 6 | 12 | 24 | 60 |
|---|---|---|---|---|---|---|---|---|---|---|
| | Feeding (breast feeding, solid food introduction, diversity, eating behaviour; SUNRISE Questionnaire) | | | | | | C | C | C | C |
| | Physical activity (M:GPAQ and Sitting Time Questionnaire) | M | M | M | M/F | | | C/M | **M** | M |
| | Step test | | M | | F | | | | | |
| | Accelerometry (substudy) | | | | | | C | | C | C |
| | Screen Time Questionnaire | M | M | M | M/F | | C | C | C | C |
| | Sleep: Pittsburgh Sleep Quality Index | M | M | M | M/F | | M | M | M | M |
| | Brief Infant Sleep Questionnaire | | | | | | C | C | C | |
| | SUNRISE sleep questionnaire | | | | | | | | | C |
| | Tobacco, alcohol, and drug use: Exposure questionnaire (WHO-STEPS, AUDIT-Questionnaire) | M | M | M | M/F | | | M | M | M |
| | Environmental tobacco exposure | | | | M | | | C | C | C |
| | Child Care Questionnaire | | | | | | M | M | M | M |
| Mental and physical health, and child development | Depression (PHQ9) | M | M | M | M/F | | | M | | M |
| | Depression (Edinburgh) | | | M | M | | M | | | |
| | Stress (PSS) | | | M | M/F | | M | | M | M |
| | Stressful life events (ACEs) | M | | | F | | | | | |
| | Generalised Self-Efficacy Scale | M | M | M | M/F | | | M | | M |
| | Social Support Questionnaire | M | M | M | M/F | | | | | |
| | Social Provisions Scale | | | | M | | | | M | M |
| | Medical history (and TB), family history, drug, supplement history | M | M | M | M/F | | M | M | M | M |
| | Breastfeeding efficacy | | | | M | | | | | |
| | Anxiety (GAD-7) | M | M | M | M/F | | | M | M | M |
| | Medication/supplement use | M | M | M | M/F | M/C | M/C | M/C | M/C | M/C |
| | Medical events (hospitalisation) | M | | M | M | M/C | M/C | M/C | M/C | M/C |
| | Births, pregnancy and reproductive health | M | | M | | M | M | M | M | M |
| | Vaccinations | | | | | | C | C | C | C |
| | Emotional health Questionnaire | M | M | M | M/F | | C | C | C | C |
| | Ages and Stages Questionnaire | | | | | | | | M | |
| | Parenting practices (Brigance Parent–Child Interaction Scale) | | | | | | | M | M | C |
| | Neurodevelopment (WHO Global Scale for Early Development—short-form) | | | | | | M | C | C | M |

**Table 2** Continued

| Domain | Measure | Preconception | | Pregnancy (week) | | Delivery | Postnatal period (months) | | | |
|---|---|---|---|---|---|---|---|---|---|---|
| Time point (months) | | Baseline | 18 months (exit) | 10–17 | 24–28 | Delivery | 6 | 12 | 24 | 60 |
| | Executive function (Early Years Toolbox) | | | | | | | | | C |
| | Temperament/behaviour (Strengths and Difficulties Questionnaire) | | | | | | | | C | C |
| | School readiness (International Development and Early Learning Assessment) | | | | | | | | | C |
| | Stimulation in the home (UNICEF Multiple Indicators Cluster Survey Early Childhood Development, Home Learning Environment) | | | | | | | | C | C |

ACE, Adverse Childhood Events; C, infant/child; DXA, dual-energy X-ray absorptiometry; F, father; M, mother.

and indicators of programme delivery in the intervention group. Sensitivity analyses will be performed to assess the influence of attrition and compliance. Intention-to-treat analysis will be used for key outcomes based on all viable pregnancies adjusted for compliance and selection of viable pregnancies to minimise bias. For the primary outcome, we will assess the effect of the intervention on FMI at age 5 years. The concept of this multiple-phase trial is that the outcomes of interest in a particular phase result from the intervention administered in the same phase, but is also mediated by the outcomes observed in the previous phase(s). For example, a reduction in obesity risks during pregnancy caused by the intervention in this phase will impact birth weight and the child weight/BMI observed in the next phase. Therefore, at the end of the study, we will measure an 'overall' effect of the interventions over the four phases on the final outcomes, but also, could assess how the association between the final outcomes and the interventions is mediated by the phase-specific outcomes (see figure 3). Other secondary anthropometric outcomes to be assessed at 5 years of age include OWO, BMI and centiles and adiposity (skin-fold thickness, arm circumference). We will also assess the effect of the intervention on glucose metabolism, blood pressure and cognitive outcomes. Mixed models with random intercept/slope effects and fixed effects (for the intervention) will be used to estimate the intervention effect. Due to possible heterogeneity of effects for key outcomes according to the time of initiation of the intervention (eg, preconception vs pregnancy), we will conduct a subgroup analysis stratifying by this variable, testing for heterogeneity. The distribution of modifiable risk factors for childhood obesity, poor cardiometabolic and neurodevelopmental outcomes at baseline and their changes over time will be assessed according to the study group.

## Governance and monitoring

The shared governance and harmonisation structures for the HeLTI trials include: Research Committee of country Principal Investigators; a single Data Monitoring Committee that is independent from the sponsor and competing interests (DMC Charter is available on the HeLTI website); WHO Monitoring and Evaluation Team; and a HeLTI Secretariat. The trial is blinded to the investigators and research data collection team, but the data management team will unblind participant's allocation on request by the DMC or ethics committee for key reporting or in cases of risk. A governance framework has been developed and adopted by all partners that details the management and access of data and biological samples. The HeLTI website (https://helti-net.org) provides more information including contact details of the HeLTI office and secretariat and information on the Data Access Portal through which investigators from around the world can view and request access to data/biospecimens from individual HeLTI country datasets. HeLTI will also be registered as a hub on Global Health

**Table 3** Longitudinal biological sample measures

| | Preconception | | Pregnancy (weeks) | | Postnatal period (months) | | |
|---|---|---|---|---|---|---|---|
| | Baseline | 18 months | 10–17 | 24–28 | 12 | 24 | 60 |
| **Mother** | | | | | | | |
| Blood collection | X (random) | X (random) | X (random) | X (fasting; OGTT) | X (fasting) | X (fasting) | X (fasting; OGTT) |
| Plasma | X | X | X | X | X | X | X |
| Serum | X | X | X | X | X | X | X |
| DNA | X | | X | X | X | | X |
| Urine (spot sample) | X | X | | X | | | X |
| Urine (dipstick) | X | X | X | X | | | X |
| **Father** | | | | | | | |
| Blood—random | | | | X | | | |
| Plasma | | | | X | | | |
| Serum | | | | X | | | |
| DNA | | | | X | | | |
| Urine (spot sample) | | | | X | | | |
| Urine (dipstick) | | | | X | | | |
| **Infant/child** | | | | | | | |
| Blood collection (DBS) | | | | | X | | |
| Blood collection | | | | | | X (fasting) | X (fasting; OGTT) |
| Plasma | | | | | | X | X |
| Serum | | | | | | X | X |
| DNA | | | | | | | X |
| Buccal swab | | | | | | X | |
| Urine (spot sample and dipstick) | | | | | | X | X |

DBS, Dried Blood Spot; OGTT, Oral Glucose Tolerance Test.

Network (https://tghn.org). The Research Committee is supported by seven cross-consortium active working groups, including: (1) Core Variable Harmonisation, (2) Interventions and Retention, (3) Early Child Development Measures, (4) Biospecimen Collection and Management, (5) Data Management and Quality Assurance, (6) Capacity Building and (7) Publications. Results will actively be disseminated within the four countries and through WHO mechanisms. An independent clinical research associate will audit the trial annually. WHO aims to regularly review quality assurance data of the trial. An electronic management system is in place for trial HHs and staff to record and manage adverse events. Utilising REDCap, Adverse Event Forms will be completed online, which will trigger a process of alerts and recommendations from the investigators, researchers and project coordinators to best handle the event. All events are also discussed at the HH weekly debrief session and reported to the DMC.

## Trial status and impact of the COVID-19 pandemic

Trial formative research started in April 2017, the pilot trial was initiated in 2018, main trial recruitment began in October 2019 and is expected to conclude early in 2022,

and the trial is expected to be completed by 2029. The collective HeLTI initiative provides an unprecedented preconception health platform with the potential to expand to other settings. To maximise the learning and impact of these studies, HeLTI aims to host postgraduate students and training opportunities, and facilitate accelerated managed access to data and biological samples. Our formative work has revealed just how complex both the environments and lives of young urban women in South Africa are. The health implications are further compounded by poorer than anticipated health literacy levels. Among these women, health is not prioritised amidst competing economic and food security concerns, and the daily stress of navigating challenging household (relationships, households with multifamilies) and community circumstances (poverty, crime and exposure to violence). Also, the Soweto environment is becoming more obesogenic with easier access to less healthy fast foods that are affordable and sold on most street corners. This makes the decision to purchase healthier, more expensive foods that are sold further away less appealing.[20] For these reasons, we have opted for a more pragmatic and adaptive trial design, whereby the intervention package can be

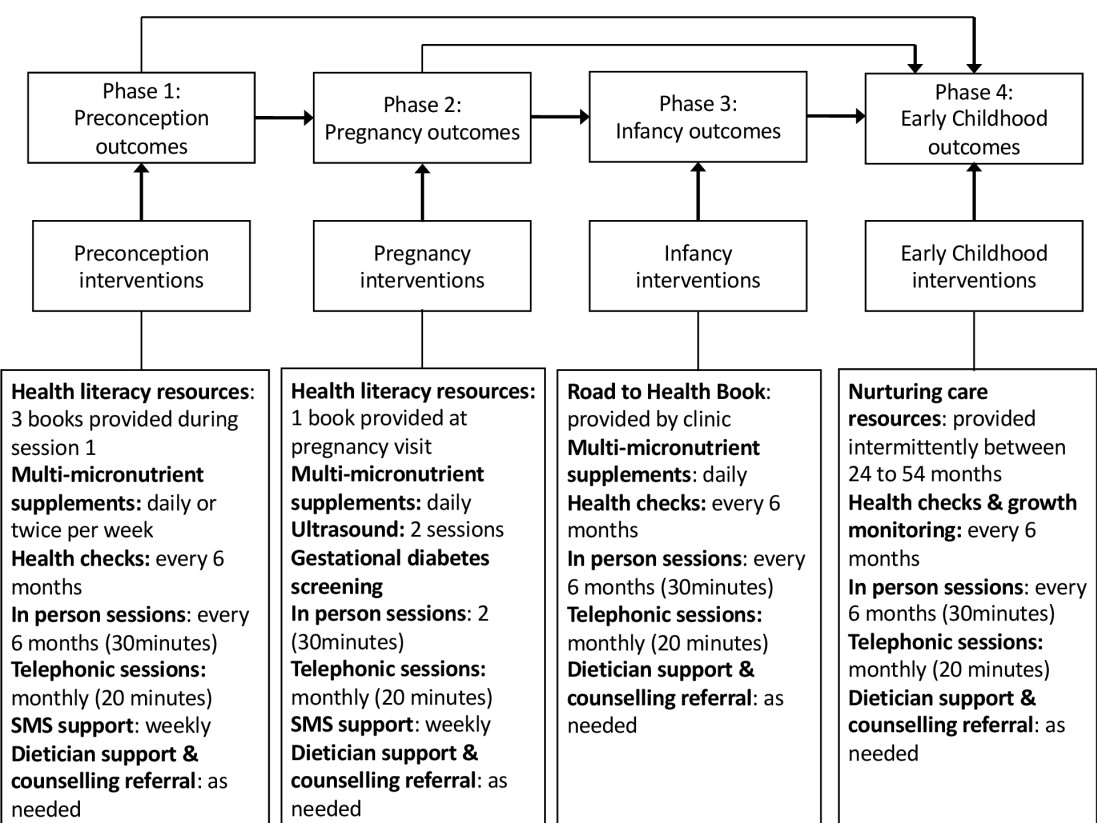

**Figure 3** Impact of outcomes in a multiphase trial and intervention components. CHWs, community health workers.

modified to the circumstances of each participant and can incorporate new learning on how best to engage, support and improve compliance. The strength of this approach is to remain relevant and responsive to the urban poor context, and thereby, increase the likelihood of achieving health behaviour change. The challenge of this adaptive and individually responsive approach is how to capture and analyse the dose effect of intervention elements. To address this challenge, we have setup a sophisticated data management and dashboard system which can monitor key quality assurance, delivery and compliance indicators that are not only useful for trial management, but also assist the intervention and control arm teams.

Six months into the main trial the COVID-19 pandemic hit South Africa through a series of waves and lockdown levels. A hard lockdown (level 5) was implemented in South Africa from 26 March to 30 April 2020. Thereafter a level 4 lockdown was implemented until 31 May 2020. Level 3 then lasted until 17 August, when the country moved to level 2 and finally level 1 (September 2021). The recruitment has been severely interrupted particularly between 26 March and 30 July 2020. On resuming community recruitment, additional COVID-19-specific questions were added to the questionnaire, support and surveillance protocols were implemented and a dedicated COVID-19 nurse was appointed. Intervention face-to-face interactions were replaced with virtual face-to-face sessions during more severe lockdowns and micronutrient supplements were delivered to participants' homes. To boost trial recruitment, we launched a revised community drive

that included community pop-up stands, and enlisted community leaders aiding recruitment drives. We have also developed a participant retention SOPs. In conclusion, changing the course of the obesity tide globally will require a major concerted effort to examine initiatives across the lifecourse. The preconception period, in particular, represents an exciting and largely untested stage in life to prevent the transgenerational impacts of maternal metabolic and mental health.

**Author affiliations**
[1]SAMRC Developmental Pathways for Health Research Unit, Department of Paediatrics, University of the Witwatersrand, Johannesburg-Braamfontein, South Africa
[2]Global Health Research Institute, School of Human Development and Health, University of Southampton, Southampton, UK
[3]Centre of Excellence of Nutrition, North-West University, Potchefstroom, South Africa
[4]DSI-NRF Centre of Excellence in Human Development, University of the Witwatersrand, Johannesburg, South Africa
[5]Lawrence S. Bloomberg Faculty of Nursing, University of Toronto, Toronton, Ontario, Canada
[6]Department of Molecular Genetics, Ontario Institute for Cancer Research, University of Toronto, Toronto, Ontario, Canada
[7]Centre for Global Child Health, SickKids Research Institute, Toronto, Ontario, Canada
[8]Institute for Global Health and Development, Aga Khan University, Karachi, Pakistan
[9]Lunenfeld-Tanenbaum Research Institute, Toronto, Ontario, Canada
[10]Medicine, Division of Infectious Diseases, Ottawa Hospital General Campus, Ottawa, Ontario, Canada
[11]Ottawa Hospital Research Institute, Ottawa, Ontario, Canada
[12]School of Public Health, University of the Witwatersrand, Johannesburg, South Africa

[13]Factor-Inwentash Faculty of Social Work, University of Toronto, Toronto, Ontario, Canada

[14]Division of Molecular Biology and Human Genetics, University of Stellenbosch, Stellenbosch, South Africa

[15]Division of Immunology, Institute of Infectious Disease and Molecular Medicine, University of Cape Town, Rondebosch, South Africa

[16]Department of Paediatrics, University of Toronto, Toronto, Ontario, Canada

[17]Community Services, Red River College, Winnipeg, Manitoba, Canada

[18]Department of Applied Psychology and Human Development, University of Toronto, Toronto, Ontario, Canada

[19]SAMRC Rural Public Health and Health Transitions Research Unit (Agincourt), Uiversity of the Witwatersrand, Johannesburg, South Africa

[20]Non-communicable Diseases Research Unit, South African Medical Research Council, Cape Town, South Africa

[21]Division of Exercise Science and Sports Medicine, University of Cape Town, Rondebosch, South Africa

[22]Chronic Diseases Initiative for Africa, University of Cape Town, Cape Town, South Africa

[23]Sydney Brenner Institute for Molecular Bioscience, University of the Witwatersrand, Johannesburg, South Africa

[24]The Hospital for Sick Children, Toronto, Ontario, Canada

[25]Department of Molecular Genetics, University of Toronto, Toronto, Ontario, Canada

[26]Department of Nutritional Sciences, University of Toronto, Toronto, Ontario, Canada

[27]Department of Paediatrics, University of the Witwatersrand, Johannesburg-Braamfontein, South Africa

[28]Department of Biochemistry and Biomedical Sciences, McMaster University, Hamilton, Ontario, Canada

[29]Department of Pharmacology and Therapeutics, McGill University, Montreal, Quebec, Canada

[30]Institute for Life Course Health Research, University of Stellenbosch, Cape Town, South Africa

[31]Department of Paediatrics, Alberta Children's Hospital Research Institute, Calgary, Alberta, Canada

[32]Department of Physiology, Obstetrics & Gynaecology and Medicine, University of Toronto, Toronto, Ontario, Canada

**Acknowledgements** The authors extend our gratitude to the South African Medical Research Council and the Canadian Institutes of Health Research for funding Bukhali. We greatly appreciate the willingness of all women who have agreed to participate in the study and the staff who are implementing this ambitious research programme. We also acknowledge the role of WHO in supporting the HeLTI collaboration and providing quality assurance support to Bukhali. We thank the HeLTI PI Research Committee and HeLTI Office for their intellectual contributions, comradery and administrative support.

**Contributors** SAN and SL are coprincipal investigators for Healthy Life Trajectory Initiative South Africa. SAN, CD, LJW, AP and SL wrote the initial protocol draft. CS, CD, PA, DB, ZB, LB, BC, TC, BF, CG, JH, JJ, HJ, JJ, KK, AK, EVL, NL, M-CM, MR, DR, SS, DS, WS, DS, MS, ST, MT, ST, SM and LR read and contributed to the final version. All authors provided edits and critiqued the manuscript for intellectual content.

**Funding** The authors extend our gratitude to the South African Medical Research Council (Strategic Health Innovation Partnerships grant) and the Canadian Institutes of Health Research for funding (grant number: HLS 151553) Bukhali. Trial Sponsor: University of the Witwatersrand, Johannesburg, South Africa; Contact Dr Drennan ( robin.drennan@wits.ac.za).

**Competing interests** None declared.

**Patient and public involvement** Patients and/or the public were involved in the design, or conduct, or reporting or dissemination plans of this research. Refer to the Methods section for further details.

**Patient consent for publication** Not applicable.

**Provenance and peer review** Not commissioned; externally peer reviewed.

**ORCID iDs**
Shane A Norris http://orcid.org/0000-0001-7124-3788
Lisa Jayne Ware http://orcid.org/0000-0002-9762-4017
CindyLee Dennis http://orcid.org/0000-0002-0135-7242
Zulfiqar Bhutta http://orcid.org/0000-0003-0637-599X
Jill Hamilton http://orcid.org/0000-0002-1958-2800
Wiedaad Slemming http://orcid.org/0000-0002-1566-8228
Mark Tomlinson http://orcid.org/0000-0001-5846-3444

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
