## [Reviewer comments · BMJ Open]

ARTICLE DETAILS

TITLE (PROVISIONAL)	Building knowledge, optimising physical and mental health, and setting up healthier life trajectories in South African women (Bukhali): a preconception randomised control trial part of the Healthy Life Trajectories Initiative (HeLTI)
AUTHORS	Norris, Shane; Draper, Catherine; Prioreshi, Alessandra; Smuts, CM; Ware, Lisa; Dennis, CindyLee; Awadalla, Philip; Bassani, D; Bhutta, Zulfiqar; Briollais, Laurent; Cameron, D; Chirwa, Tobias; Fallon, B; Gray, CM; Hamilton, Jill; Jamison, J; Jaspan, Heather; Jenkins, Jennifer; Kahn, Kathleen; Kengne, AP; Lambert, Estelle; Levitt, Naomi; Martin, Marie-Claude; Ramsay, Michele; Roth, Daniel; Scherer, Stephen; Sellen, Daniel; Slemming, Wiedaad; Sloboda, Deborah; Szyf, M.; Tollman, Stephen; Tomlinson, Mark; Tough, Suzanne; Matthews, Stephen; Richter, Linda; Lye, Stephen

VERSION 1 – REVIEW

REVIEWER	Kipo-Sunyehzi, Daniel University of Ghana, Legon Centre for International Affairs and Diplomacy
REVIEW RETURNED	27-Dec-2021

GENERAL COMMENTS	REVIEWER COMMENTS ON study protocol for BMJ Open bmjopen-2021-059914 "Building knowledge, optimising physical and mental health, and setting up healthier life trajectories in South African women (Bukhali): a preconception randomised control trial part of the Healthy Life Trajectories Initiative (HeLTI)" The purpose of the research and study setting are clear from the introduction. It is an interesting study. This issue needs to be addressed by the authors on Article Summary Introduction: 1) It is not clear why the location(s) in South Africa such as the Soweto, an urban-poor area of Johannesburg targeting these women aged 18-28 years. The authors should provide a justification for the selection of the location(s). On the background, it is indeed a worrying trend in non-communicable diseases (NCDs) like obesity, hypertension, and diabetes at the global level and sub-Saharan Africa in particular in terms of the number of persons who died as a results of NCDs. The first two paragraphs rigor in terms of literature seem to be limited and hope some additional information may be added.
--

	Methods/design Participant and public involvement, the multi-actor approach (Participant Advisory Group and Stakeholder Advisory Group) and the extensive social media engagements are very insightful. The study design is very interesting, appropriate and innovative with the Bukhali individual randomised controlled two-arm trial, each participant as the unit of randomisation (1:1 ratio).. On intervention: community health worker approach, these three theories are mentioned and not developed in the paper. They are the theory of planned behaviour, control theory and social cognitive theory. It will be good the authors expand on them and connect them to the research.
--	---

REVIEWER	Abdulahi, Misra Jimma University College of Public Health and Medical Sciences, Population and Family Health
REVIEW RETURNED	03-Jan-2022

GENERAL COMMENTS	This is a well-designed RCT aimed at addressing an important public health topic: Childhood obesity. My concerns are presented below. Participants What is the rationale to recruit women 18-28 years of age? Are there additional possible exclusion criteria that can be applied at different phases of the intervention and data collection time points? Randomization The type of randomization used to randomize study participants to either of the group is not mentioned. Allocation concealment It is not clear how allocation concealment was ensured until the intervention is assigned. Blinding Except mentioning that the researcher and research team collecting trial data will be blinded to participant allocation throughout the trial, how blinding will be implemented is not stated. The intervention The intervention described on page 10 states that there are four different phases (preconception, pregnancy, infancy and childhood). However, according to figure 3, the intervention is planned in three phases (preconception, pregnancy and early childhood). I suggest the authors be consistent about the phases of the intervention throughout the document. The 'how' and 'when' of the intervention is not clearly stated. For instance, the health literacy component of the intervention is planned to be provided thrice during preconception, twice during pregnancy and once during early childhood. But, when are the specific schedules for the three contacts during preconception? How and where do these contacts take place? What will be the estimated duration of each contact? Intervention providers What is the rationale to use CHW as intervention providers? What are the eligibility criteria to select them? Evaluation of the intervention The authors have designed a complex intervention following the necessary steps of developing, piloting and evaluation. As part of the evaluation, they have also planned to do a process evaluation focusing focus on implementation, mechanisms of impact and
--

	context. However, including an economic evaluation would make the results of the trial useful for decision-making. Do the authors have a plan to do an economic evaluation? Participant timeline It is recommended to include the time schedule of enrolment, interventions, assessments, and visits for participants. The authors have included assessments and visits for participants in table 2. The authors may choose to modify this table or prepare a new one to include the time of enrolment, the study period (enrolment, allocation, post-allocation and close-out) and the duration of each intervention. Participant flow It is strongly recommended to present a participant flow diagram.
--	--

VERSION 1 – AUTHOR RESPONSE

Reviewer: 1

Dr. Daniel Kipo-Sunyehzi, University of Ghana

On a whole, it is a very good paper and I recommend ACCEPT WITH MINOR REVISION.

We appreciate your review and your suggestions

1. It is not clear why the location(s) in South Africa such as the Soweto, an urban-poor area of Johannesburg targeting these women aged 18-28 years. The authors should provide a justification for the selection of the location(s).

We included the reasoning for why Soweto (community need, high burden of obesity, infrastructure and expertise) in the background section, and the reasoning for the age range (significant proportion of young Soweto women are already overweight or obese by the time they have their first child) in the methods section.

2. On the background, it is indeed a worrying trend in non-communicable diseases (NCDs) like obesity, hypertension, and diabetes at the global level and sub-Saharan Africa in particular in terms of the number of persons who died as a results of NCDs. The first two paragraphs rigor in terms of literature seem to be limited and hope some additional information may be added.

We removed some superfluous sections and added some information to strengthen the rationale that gave birth to this RCT.

3. On intervention: community health worker approach, these three theories are mentioned and not developed in the paper. They are the theory of planned behaviour, control theory and social cognitive theory. It will be good the authors expand on them and connect them to the research.

We have succinctly explained these theories and how they formed the basis for the behaviour change model in the community health worker section.

Added section:

Using the taxonomy of behaviour change techniques (BCTs)³³, we grounded the behaviour change intervention in: (i) theory of planned behaviour, which suggests that behaviours are determined by attitudes, subjective norms and perceived control; (ii) control theory, which suggests we use inner- and outer-control to minimise deviation from the standard; and (iii) social cognitive theory describes that behaviour and choices are influenced by the environment, personality and factors such as goals. These theories combined anchors the intervention to value the importance of context, individual needs, and support to bolster agency.

4. Healthy Conversation Skills (HCS) is another innovative and the training given to health helpers to assist the socioeconomically disadvantaged participants with the aim to help them improve self-

efficacy and support behaviour change in a way to help people including disadvantaged in society to find solutions to challenges they encounter. This part is well explained though with no sources (citations). This is a very good research project paper.

Thank you, we added references at the end of the paragraph but recognise that it may be easily overlooked, therefore, we have shifted these to the beginning.

Reviewer: 2

Dr. Misra Abdulahi, Jimma University College of Public Health and Medical Sciences

Comments to the Author:

This is a well-designed RCT aimed at addressing an important public health topic: Childhood obesity. My concerns are presented below.

Many thanks for your review we appreciate your comments.

1. What is the rationale to recruit women 18-28 years of age?

We included the reasoning for the age range (significant proportion of young Soweto women are already overweight or obese by the time they have their first child) in the methods section.

Modified section:

...we identified our young women (18-28 years) as the target group given that a high proportion these women are already either overweight or obese and most will have their first child during this age range...

2. Are there additional possible exclusion criteria that can be applied at different phases of the intervention and data collection time points?

We aimed to place minimal restrictions for continued participation to mimic "real-world" scenario as if this was implemented through the primary health care system. Therefore, there are no further exclusion criteria.

3. The type of randomization used to randomize study participants to either of the group is not mentioned. Allocation concealment It is not clear how allocation concealment was ensured until the intervention is assigned.

Thank you for pointing this out. We have added more detail.

Modified section:

Bukhali is an individual-randomised controlled two-arm trial, (simple randomisation; 1:1 ratio to the intervention arm or to standard of care plus control arm). Only after informed consent and baseline data collection, will the participant move to a separate/private randomisation counter where the participant clicks on a computer-generated electronic envelope that reveals the allocation to the participant and the randomisation officer. The randomisation officer then escorts the participant to a separate wing on a different floor in the centre to introduce to either the intervention arm or control arm coordinator, thus preserving the concealment of allocation. The research team collecting trial data and associated researchers will be blinded to participant allocation throughout the trial, this will be achieved by: (i) ensuring that group allocation is locked on the database and cannot be accessed by the research team; (ii) training the research team not to ask and participants will be encouraged not to divulge their allocation status; and (iii) when participants visit the centre for their respective arm programme these activities will occur in a separate part of the centre away from the research team.

4. Except mentioning that the researcher and research team collecting trial data will be blinded to participant allocation throughout the trial, how blinding will be implemented is not stated.

This has been expanded, please see above.

5. The intervention described on page 10 states that there are four different phases (preconception, pregnancy, infancy and childhood). However, according to figure 3, the intervention is planned in three phases (preconception, pregnancy and early childhood).

I suggest the authors be consistent about the phases of the intervention throughout the document. The 'how' and 'when' of the intervention is not clearly stated. For instance, the health literacy component of the intervention is planned to be provided thrice during preconception, twice during pregnancy and once during early childhood. But, when are the specific schedules for the three contacts during preconception? How and where do these contacts take place? What will be the estimated duration of each contact?

We acknowledge that this is confusing and inconsistent. We have simplified throughout the paper and inserted a patient flow diagram (Fig.1), replaced Fig. 2, and expanded Fig. 3 with summary information on the phases, analysis and intervention components.

6. What is the rationale to use CHW as intervention providers? What are the eligibility criteria to select them?

Thank you for highlighting this. We have added more information.

Modified section:

We opted for the CHW approach as there is strong efficacy evidence in Africa, and it aligns with SA's primary health care re-engineering policy. The HHs will share similar qualifications, characteristics and salary levels to CHWs in SA, and eligibility includes: women between the ages of 23-40 years with completed secondary school with some tertiary education, but no formal training as a CHW is required.

7. The authors have designed a complex intervention following the necessary steps of developing, piloting and evaluation. As part of the evaluation, they have also planned to do a process evaluation focusing focus on implementation, mechanisms of impact and context. However, including an economic evaluation would make the results of the trial useful for decision-making. Do the authors have a plan to do an economic evaluation?

Currently, we do not have an economic evaluation protocol but aim to work with health economists to develop one. We have listed this as a limitation.

8. It is recommended to include the time schedule of enrolment, interventions, assessments, and visits for participants. The authors have included assessments and visits for participants in table 2. The authors may choose to modify this table or prepare a new one to include the time of enrolment, the study period (enrolment, allocation, post-allocation and close-out) and the duration of each intervention. It is strongly recommended to present a participant flow diagram.

We have included a participant flow figure (Fig.1) and incorporated several of your suggestions into Fig. 3.

VERSION 2 – REVIEW

REVIEWER	Abdulahi, Misra Jimma University College of Public Health and Medical Sciences, Population and Family Health
REVIEW RETURNED	16-Feb-2022

GENERAL COMMENTS	I recommend accepting the manuscript.
---------------------------------------